# Lymphatic filarial serum proteome profiling for identification and characterization of diagnostic biomarkers

Vipin Kumar[1], Ayushi Mishra[1], Awadehesh Kumar Yadav[2], Sushma Rathaur[1], Anchal Singh[1]*

1 Dept. of Biochemistry, Institute of Science, Banaras Hindu University, Varanasi, Uttar Pradesh, India,
2 National Centre for Disease Control, Ministry of Health and Family Welfare, Varanasi, Uttar Pradesh, India

* anchalsinghbhu@yahoo.com, anchalsingh@bhu.ac.in

**Data Availability Statement:** All relevant data are within the paper and its Supporting information files.

## Abstract

Lymphatic Filariasis (LF) affects more than 863 million people in tropical and subtropical areas of the world, causing high morbidity and long illnesses leading to social exclusion and loss of wages. A combination of drugs Ivermectin, Diethylcarbamazine citrate and Albendazole is recommended by WHO to accelerate the Global Programme to Eliminate Lymphatic Filariasis (GPELF). To assess the outcome of GPELF, to re-evaluate and to formulate further strategies there is an imperative need for high quality diagnostic markers. This study was undertaken to identify Lymphatic Filarial biomarkers which can detect LF infections in asymptomatic cases and would also serve as indicators for differentiating among different clinical stages of the disease. A combination of Fourier-transform infrared spectroscopy (FT-IR), MMP zymography, SDS-PAGE, classical 2DE along with MALDI-TOF/MS was done to identify LF biomarkers from serum samples of different stages of LF patients. FT-IR spectroscopy coupled with univariate and multivariate analysis of LF serum samples, revealed significant differences in peak intensity at 3300, 2950, 1645, 1540 and 1448 cm$^{-1}$ (p<0.05). The proteomics analysis results showed that various proteins were differentially expressed (p<0.05), including C-reactive protein, α-1-antitrypsin, heterogeneous nuclear ribonucleoprotein D like, apolipoproteins A-I and A-IV in different LF clinical stages. Functional pathway analysis suggested the involvement of differentially expressed proteins in vital physiological pathways like acute phase response, hemostasis, complement and coagulation cascades. Furthermore, the differentiation between different stages of LF cases and biomarkers identified in this study clearly demonstrates the potential of the human serum profiling approach for LF detection. To our knowledge, this is the first report of comparative human serum profiling in different categories of LF patients.

## Introduction

Lymphatic Filariasis (LF) is one of the most prevalent tropical diseases affecting more than 863 million people in tropical and subtropical areas of the world (https://www.who.int/news-

**Funding:** The author(s) received no specific funding for this work.

**Competing interests:** The authors have declared that no competing interests exist.

**Abbreviations:** AAT, Alpha-1 antitrypsin; APPs, Acute phase proteins; CRP, C—reactive protein; DAVID, Database for annotation, visualization and integrated discovery; DEP, Differentially expressed protein; DTAG, Diagnostic technical advisory group; FT-IR, Fourier-transform infrared spectroscopy; GPELF, Global Programme to Eliminate Lymphatic Filariasis; hnRNP-D, heterogeneous nuclear ribonucleoprotein-D; IFF, Immunoglobulin free fraction; LF, Lymphatic filariasis; MALDI-TOF/MS, Matrix-assisted laser desorption/ionization-time of flight mass spectrometry; MMP, Matrix metalloproteinase; PANTHER, Protein annotation through evolutionary relationship; PCA, Principal component analysis; PTBP1, Polypyrimidine tract- binding protein 1; SAA, Serum amyloid A; STAT4, Signal transducer and activator of transcription 4; TIMPs, Tissue inhibitors of metalloproteinases.

room/fact-sheets/detail/lymphatic-filariasis). The disease is considered a major obstacle to socioeconomic development in endemic countries and is a major reason for permanent and long-term disabilities worldwide. LF infection is caused by three nematode worms *Wuchereria bancrofti*, *Brugia malayi* and *Brugia timori* [1]. These worms reside in the hosts' lymphatic system for several years obstructing the hosts' lymphatic flow resulting in extensive morbidity because of pathologies like hydrocele, lymphedema and elephantiasis. To eradicate this debilitating disease, in the year 2000, World Health Organization (WHO) launched the Global Programme to Eliminate Lymphatic Filariasis (GPELF), which aims for the global eradication of LF [2]. Lately, WHO is recommending an annual dose of drugs Ivermectin, Diethylcarbamazine citrate and Albendazole (IDA) to accelerate the global elimination of Lymphatic Filariasis [3]. IDA treatment will not only interrupt filarial transmission but would also reduce morbidity and mortality associated with LF. IDA coverage has to be provided for several years to the entire population at the risk of infection so that LF transmission rates are substantially lowered.

Though, LF eradication by the year 2030 is being projected, however, most of the current LF diagnostic tests are suffering from one or more drawbacks [4]. The most popular diagnostic test "night blood film examination" can provide evidence of active infection needed for surveillance activities. However, the test suffers from a number of limitations and usually, patients show resistance as samples should only be collected during the night. Species- specific antigens such as Wb 123 antigen and a few antifilarial antibody detection tests have been developed based on the use of recombinant antigens e.g. Wbsxp1 and WB123 from *W. bancrofti* and another Bm14 from *B. malayi* [5], nevertheless, issues related with cross-reactivity have been frequently reported. Wb123 can be primarily used in untreated endemic populations, yet further evaluations and guidelines will be required to define its use in populations that have undergone treatment for the control of LF [6]. WHO has recommended Immuno-chromatographic card test (ICT) for LF diagnosis, which is based on detection of anti-filarial antibody. However, since its initial trials ICT has been facing challenges about its sensitivity, production, storage, cross-reactivity as well as cost-effectiveness for field trials [7]. Further, the available LF immunoassay tests are unable to provide the infection load and microfilaricidal efficacy following drug administration.

Monitoring of filarial infection in a population undergoing IDA is strongly recommended to assess the outcome of GPELF and to re-evaluate and formulate further strategies for success of GPELF. The situation becomes more complicated due to lack of LF biomarkers for active infection leading to difficulties in diagnosing active infection and in assessing the therapeutic success of IDA. Sensitive diagnostic markers are urgently needed in places/regions where transmission of infection is low and for mapping prevalence in hypo-endemic areas.

Under disease conditions, various serum components like proteins, carbohydrates, and lipids undergo rapid alterations which can be detected by serum profiling. Human serum profiling is an invaluable source for the identification of disease-related markers, disease pathogenesis and host immune response. Lately, serum profiling has been done to investigate disease induced alterations in infectious diseases like hepatitis [8], leishmaniasis [9], malaria [10], leptospirosis [11] and also in different cancers [12, 13]. Some of these studies have used FT-IR spectroscopy coupled with multivariate analysis whereas some have entirely focused on proteomics analysis.

The present study was undertaken to identify LF biomarkers from the human population that can be used for identifying active LF infection and for discriminating between different clinical stages of LF. Lymphatic Filariasis is clinically categorized into three stages Asymptomatic or Stage I, Acute or Stage II and Chronic or stage III depending upon the disease progression. A majority of LF infections are asymptomatic and most individuals have down-regulated

immune response as well as their lymphatic and /or renal systems might be damaged to some extent. Stage II or Acute infections mostly involve chills, fevers, pain and inflammation of limbs/genitals which may eventually start developing into lymphedema and elephantiasis. Stage III or chronic manifestations of LF include lymphedema in breasts and lower limbs or genitals. Often the affected tissue hardens and folds allowing bacterial and fungal growth culminating in secondary infections.

The most common LF diagnostic methods are ICT (immune-chromatographic) Test, Lymphoscintinography, X-ray diagnosis and Ultrasound but these methods have limited sensitivity, high cost and cumbersome sample processing. The Asymptomatic patients have no external signs of LF infection and hence remain undiagnosed till chronic symptoms appear. The popular night blood film examination does not show positive result if microfilariae load is low. In this study, we have used a combination of Fourier Transform Infrared (FT-IR) spectroscopy, MMP zymography, SDS-PAGE, classical 2DE along with MALDI-TOF/MS [14] to identify biomarkers from serum samples of LF patients. FT-IR spectroscopy coupled with univariate and multivariate analysis revealed significant spectral differences of LF serum samples with healthy controls whereas the proteomics analysis results showed differential expression of several proteins which have not been reported in Lymphatic Filariasis previously. Although the serum comparative proteome analysis can be a bit expensive but this study can pave the way for developing potential biomarkers for LF infections. In the future, the laboratory is planning to select one or two biomarkers from the proteomic studies which could be used for identification of different stages of LF infection using more convenient, cost effective technique like ELISA or FTIR.

## Materials and methods

### Ethical statement

This study was approved by the Institutional Ethical Committee of Banaras Hindu University (*Ref No*: *I.Sc./ECM-XII/2018-19/07 dated*: *28.04.2018)*, Varanasi. After summarizing the detailed experimental purpose in local language, written informed consent was taken from all the participants. Prior to the study, each subject was physically and clinically examined and inquired about disease status.

### Inclusion and exclusion criteria

Study participant were (i) $\geq$ 20 year of age and/or (ii) $\geq$ 40 kg body weight (iii) non pregnant women (iv) no evidence/ history of co-morbidity were included in the study. Healthy volunteers (n = 20) of age group 20–50 years living in endemic areas and free from LF infection were included as control for the study.

Study participant (i) less than 20 years and/or (ii) less than 40 kg weight (iii) pregnant women (iv) participants with severe chronic illness like liver disease, renal insufficiency, diabetes, cancer or any other illness (v) unable to provide informed consent were excluded from this study.

### Testing for active LF infection and clinical symptoms

For microscopic examination, around 20 µl of finger prick blood sample was spotted onto a glass slide and a thick smear was prepared. After staining with Leishman's stain, the slides were examined for the presence of microfilariae (mf). In total, 87 participants were selected for blood collection in which 23 were asymptomatic, 19 were acute, 25 were chronic patients and 20 were healthy individuals (Normal). The LF infected cases were examined by a clinician and

were categorized based on the above mentioned manifestations and presence/absence of microfilariae in the bloodstream.

## Collection of blood and isolation of serum

~5 ml of venous blood was taken from all the study participants, that is, from healthy volunteers and different stages of filarial patients. Serum was separated by allowing the tubes to stand at room temperature for 1 hr, followed by centrifugation at 7000g for 10 min, at 4˚C. A protease inhibitor cocktail (Sigma-Aldrich) was added to the serum samples which were later divided into aliquots and stored at -20˚C, until use [15]. Protein Estimation was done by Bradford's method and bovine serum albumin was used as a standard [16].

## Separation of immunoglobulins

For immunoglobulin separation, 1 ml of saturated ammonium sulfate was added to 0.5 ml serum samples and 0.5 ml cold PBS, stirred thoroughly at 4˚C for 30 min, followed by centrifugation at 5000g for 10 mins. After centrifugation, the precipitate was suspended in cold PBS. The precipitate was again washed twice with PBS followed by dialysis at 4˚C for 3h. Lastly, protein estimation was done in the dialyzed samples [17].

## Separation of immune complexes

Immune complexes from serum of healthy normal and LF cases were isolated by the method of Menikou et al., 2019 [18] with minor modification. Immune complexes were precipitated from the serum after overnight incubation at 4˚C with an equal volume of PBS containing 8% Polyethylene glycol. Precipitates were washed thrice with 4% PEG in PBS and then protein content was estimated.

## FT-IR analysis of serum samples

FT-IR (Fourier-transform infrared spectroscopy) analysis of serum was performed with Perkin Elmer Spectrum 65, FTIR spectrometer. Serum samples were mixed with 50 μl chilled methanol and vortexed for 30 seconds. Next, the samples were kept at -20˚C for 30 minutes followed by centrifugation at 10000 rpm, for 15 minutes at 4˚C. Pellet was collected for FT-IR spectrum recording from 4000 $cm^{-1}$ to 500 $cm^{-1}$ [19]. Signal to noise ratio of spectra was improved by 100 inferograms with a special resolution of 4 $cm^{-1}$ average. Further, background spectra were recorded and subtracted from sample spectra under identical conditions. Each sample measurement was performed in triplicates. The original FT-IR spectral files were imported in Origin Pro 8.0 for peak picking, peak integration, feature identification and labeling. Spectral quality was controlled by normalizing and removing the spectral background. Univariate and multivariate data analysis was performed by MetaboAnalyst 5.0 [20]. Fold change analysis was used for univariate analysis whereas principal component analysis (PCA) was used for multivariate data analysis.

## SDS-PAGE

40 μg serum proteins were subjected to 7.5% SDS-PAGE for protein separation. The serum samples were mixed in sample buffer (40 mM Tris-Cl pH 6.8, 10% SDS, 10% glycerol, and 0.1% bromophenol blue) [21] and subjected to non-reducing electrophoretic separation. The gel was stained with Coomassie Brilliant Blue R-250, followed by de-staining and the band intensity was analyzed by Quantity One and ImageJ software. Each experiment was repeated thrice with all samples.

## Gelatin zymography

Gelatin Zymography was performed in 7.5% SDS-PAGE having 0.1% (w/v) gelatin as the substrate. Serum samples were mixed in sample buffer (40 mM Tris-Cl pH 6.8, 10% SDS, 10% glycerol, and 0.1% bromophenol blue) and separation was carried out at 15 μA. Gels were washed in 2% Triton X100 for 45 minutes to remove SDS [22]. After washing, gels were incubated for 18 h in 40 mM Tris-Cl pH 7.5 with 0.15M NaCl, 10 mM $CaCl_2$ and 0.01% Brij 35. Coomassie Brilliant Blue R-250 staining was done and band intensity was analyzed by Quantity One and ImageJ software. The experiment was carried out in triplicate for all the samples.

## Two-dimensional gel electrophoresis (2DE)

Serum samples were treated with 10X volume ice-chilled acetone and kept at −20˚C (5 hrs) for protein precipitation followed by centrifugation at 7500 rpm for 10 min at 4˚C. Pellet was collected and rehydrated in 200 μl of the rehydration solution (7 M urea, 2 M thiourea, 2% w/v CHAPS, 15 mM DTT, 0.5% v/v IPG buffer pH 3–10). The samples were analyzed using 11 cm IPG strips with PI value 3–10 for better resolution. The isoelectric focusing (IEF) was performed using a Protean IEF Cell (BioRad, United States) at 20˚C as follows: 15min at 250V, rapid ramping to 8,000V for 2h and 8,000V for 26,000 Vh (using a limit of 50 μA/strip) for 7 hours. After IEF, the strips were first equilibrated with 40 mM Tris-HCl buffer (pH 8.8) containing 6 M urea, 25% w/v glycerol, 2% w/v SDS, 1% w/v DTT and 2.5% iodoacetamide. The second dimension was done on 10% SDS PAGE followed by staining with Coomassie Brilliant Blue G-250 (10% Aluminum sulfate, 10% ethanol, 0.02% CBB G-250, and 2.5% ortho-phosphoric acid), and images were taken with a gel documentation system (Alpha Innotech, USA) and analyzed by PDQuest software (BioRad, USA) [23]. Three independent experiments were performed to ensure the reproducibility of results. The protein spot intensity was subjected to ANOVA ($p < 0.05$) using Origin software. Protein spots showing altered expression among LF cases and normal (ratio ≤1.5) were selected for MALDI-TOF/MS analysis.

## In-gel protein digestion

The differentially expressed spots/bands were excised and de-stained by using 1:1 ratio of 15mM $K_3$ [Fe(CN)$_6$] and 50 mM $NH_4HCO_3$ for 15 minutes. Next, washing was done with 25 mM Ammonium bicarbonate and dehydration with Acetonitrile (ACN). The sample was treated with 100 mM DTT at 60˚C for 1hr followed by 250 mM Iodoacetamide at room temperature in dark for 45 minutes. Trypsin digestion was done overnight at 37˚C, then the digested peptides were extracted in 0.1% Tri-fluoro Acetic acid (TFA), dried and dissolved in 5μl of TA buffer (1:1 ratio of 0.1% TFA in water and 100% ACN) followed by MALDI-TOF/MS.

## Matrix-assisted laser desorption/ionization-time of flight mass spectrometry (MALDI-TOF/MS)

The sample was mixed with an equal volume of HCCA (α-Cyano-4-hydroxycinnamic acid) matrix (5 mg/mL HCCA 1:2 ratio of 0.1% TFA and 100% ACN) and 1.5 μl of air-dried sample was analyzed by the MALDI TOF/TOF ULTRAFLEX III instrument (Bruker Daltonics, Germany). PEPMIX mixture was used for external calibration from mass range 1046 Da to 3147 Da. Further analysis was performed with Flex Analysis Software (Version 3.3) in reflectron ion mode. The masses obtained in the MS-MS were subjected to Mascot (Version 3.3) search against Swiss-Prot "*Homo Sapiens*" database for identification of proteins. Identified proteins

having a minimum of 2 matching unique peptides were further analyzed. A confidence interval of $\geq$ 95% was maintained for protein identification.

## Protein networks and functional analysis

The differentially expressed proteins identified by MALDI-TOF/MS were subjected to the STRING tool for functional protein interaction network analysis. DAVID (Database for Annotation, Visualization, and Integrated Discovery) [24] version 6.8 and PANTHER (Protein Annotation through Evolutionary Relationship) database version 16.0 [25]. The functional annotation was done by uploading a list of Uniprot accession numbers of differentially expressed proteins and mapping with *Homo sapiens* data set [26]. The gene ontology terms, biological processes, cellular components, molecular function, protein class, and pathway analysis for each dataset were obtained from PANTHER.

## Statistical analysis

Every experiment was performed in triplicates (n = 3). Each experiment was repeated at least twice and data is expressed as mean ± SD. The statistical analysis was performed by comparing each LF clinical stage with control group using two tailed Student's t-test and one way ANOVA in Orign Pro 8.0 software. For FTIR analysis we have done Principle component and fold change analysis.

## Results

### Protein estimation in different categories of LF samples

The comparative serum analysis of LF cases showed that serum protein concentration increased following LF infections. The protein concentration in normal serum samples was 69 ± 8.0 mg/ml, 78 ± 12.0 mg/ml in asymptomatic, 81 ± 14 mg/ml in acute cases, whereas chronic cases had 93 ± 19 mg/ml (p <0.05) serum protein (Table 1). The change in average serum protein content could be directly related to LF infection and disease progression. Further, the protein content of immunoglobulin free fraction (IFF) was estimated and it was found that the protein content was elevated in all the clinical stages of LF. The asymptomatic samples had highest IFF protein content (8.11 ± 1.3 mg/ml) (p <0.05) followed by acute (8.02 ± 0.9 mg/ml), chronic (7.9 ± 0.8 mg/ml) and normal (7.9 ± 0.8 mg/ml) samples. The protein content in immune complexes of asymptomatic (21.23 ± 2.4 mg/ml) (p <0.05) and acute (18.21 ± 3.0 mg/ml) cases was elevated, however chronic samples (14.5 ± 1.5 mg/ml) had even lower protein content than normal samples (15.31 ± 2.8 mg/ml).

**Table 1. Total protein concentration in different category of LF infected human serum.**

| S. N. | Category of Serum | Serum Protein Conc.(mg/ml) | Protein Conc. in Immunoglobulin free fraction (mg/ml) | Protein Conc. in Immune Complex (mg/ml) |
|---|---|---|---|---|
| 1. | Normal | 69 ± 8.0 | 6.85 ± 1.2 | 15.31 ± 2.8 |
| 2. | Asymptomatic or stage I | 78 ± 12.0 | 8.11* ± 1.3 | 21.23* ± 2.4 |
| 3. | Acute or Stage II | 81 ± 14.0 | 8.02 ± 0.9 | 18.21 ± 3.0 |
| 4. | Chronic or Stage III | 93* ± 19.0 | 7.9 ± 0.8 | 14.5 ± 1.5 |

Data expressed in mean ± standard deviation of all LF cases and normal subjects.

*Indicate P value < 0.05 is considered as significant.

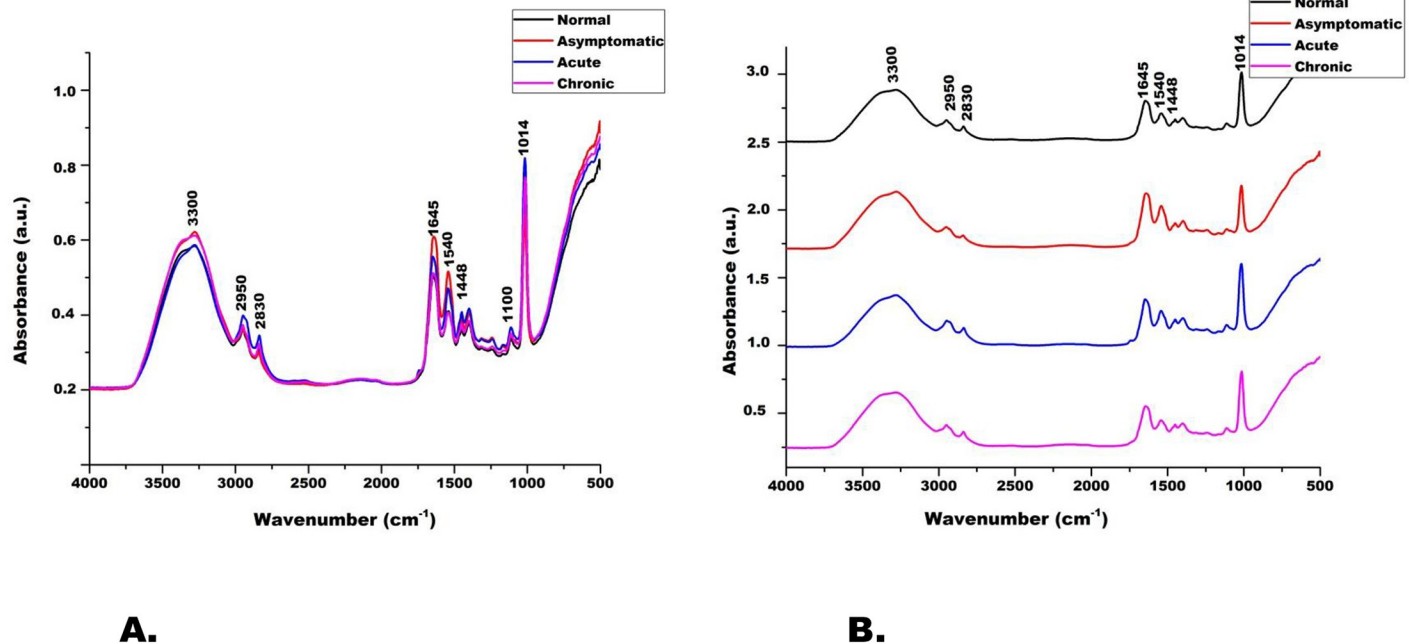

**Fig 1. Normalized average FT-IR spectra of normal and LF infected human sera in the range of 4000–400 cm⁻¹.** A.) comparative intensity analysis and B) waterfall analysis.

## FT-IR spectral analysis

FT-IR spectra of biological samples are mostly performed in 4000 cm⁻¹ to 400 cm⁻¹ to detect alterations in peaks corresponding to proteins, lipids, carbohydrates and nucleic acids. Fig 1 shows the mean normalized and baseline corrected FT-IR spectra of different categories of LF serum and control samples in the range of 4000 cm⁻¹ to 400 cm⁻¹. The preliminary data shows that the LF samples were spectrally unique in lipid and protein regions in comparison to control samples. Although, the peaks in the average FT-IR spectra of different LF cases and control serum were similar in shape however all the peaks were higher in intensity in LF samples as compared to control serum samples (Table 2). The spectral peak at 3300 cm⁻¹ corresponding to the O-H and N-H stretching of proteins and amino acids was of higher intensity in asymptomatic and chronic cases as compared to control samples. The peak at 1645 cm⁻¹ represents C = O stretching, N-H bending and C-N stretching of proteins, hence any structural alteration in the secondary structure of proteins is reflected in this peak. The peak at 1540 cm⁻¹ represents the amide II protein band and is attributed to N-H bending caused due to stretching of C-N in the protein structure. The peaks at 1645 cm⁻¹ and 1540 cm⁻¹ were significantly higher in asymptomatic cases as compared to other LF and control samples. The FT-IR peak at 2950 represents asymmetric C-H stretching of methyl ($CH_3$) groups whereas the peak at 1448 cm⁻¹ is attributed to bending of methylene groups and both of these peaks were of higher intensity in acute serum samples.

The FT-IR data was analyzed by both univariate (fold change) and multivariate (PCA) approaches. The fold change analysis of FT-IR data was used to compare the data of normal samples with different LF categories. The results of the fold change analysis are given in S1 Fig. The peaks at 3300, 2950, 1645, 1540 and 1448 cm⁻¹ were found to be significantly altered in asymptomatic and acute cases. Multivariate data analysis of FT-IR results was also done by

**Table 2. Peaks assignment for FTIR spectra of normal and LF cases.**

| S.N. | FTIR Peak (cm$^{-1}$) | Peak Assignment | Peak Specification |
|---|---|---|---|
| 1. | 1014 | C-O stretching of carbohydrates | High Intensity in Acute* and Asymptomatic |
| 2. | 1100 | PO$_2^-$ asymmetric and symmetric stretching | High Intensity in Acute and Asymptomatic* |
| 3. | 1300 | Amide III: proteins | High Intensity in Asymptomatic* and Acute |
| 4. | 1400 | COO—symmetric stretching | High Intensity in Acute* and Asymptomatic** |
| 5. | 1448 | Bending of CH$_3$ group | High Intensity in Acute** and Asymptomatic** |
| 6. | 1540 | N–H bending, C–N stretching vibrations Amide II | High Intensity in Asymptomatic** and Acute* |
| 7. | 1645 | C = O stretching, N-H bending and C-N stretching in amide I | High Intensity in Asymptomatic*** and Acute** |
| 8. | 2830 | CH2 antisymmetric stretching for lipid | High Intensity in Acute* and Asymptomatic* |
| 9. | 2950 | CH3 antisymmetric stretching: lipids, protein side chains | High Intensity in Acute** and Asymptomatic* |
| 10. | 3300 | N-H stretching of amide A | High Intensity in Asymptomatic*** and Chronic* |

P value < 0.05 is considered as significant.

***P < 0.001,

**P < 0.01,

*P< 0.05

Principal component analysis (PCA) which explains the variations in data sets. Since it is really difficult to interpret complex datasets hence PCA is used to reduce the dimensionality without informational loss. PCA analysis of FT-IR data showed that PC1 and PC2 could explain 99%, 99.3%, and 99.7% data variability of asymptomatic, acute and chronic LF cases (S2 Fig).

## 1D SDS-PAGE of serum samples

A comparative serum profiling of different categories of LF sera was done by 1D-SDS PAGE. In total 60 samples were analyzed out of which 15 were asymptomatic, 15 were acute, 15 were chronic and 15 were normal. The Coomassie-stained gels showed 4 protein bands of molecular mass 145 kDa, 93 kDa, 84.1 kDa and 65 kDa which were higher in LF samples in comparison to normal serum (Fig 2). The band intensities of healthy and LF serum were compared by Quantity one and ImageJ software. It was observed that 145 kDa band was 2.23 fold higher in asymptomatic, 2.12 fold higher in acute whereas 1.59 times higher in chronic compared to normal. The band at 93 kDa was 1.65 fold higher in asymptomatic, 1.94 fold higher in acute and 1.90 times higher in intensity in chronic LF cases compared to healthy normal subjects. The next band at 84.1 kDa was 1.21 fold higher in asymptomatic, 1.54 fold higher in acute and 1.32 fold higher in chronic cases compared with normal. The lightest band of 65 kDa was 1.63 fold higher in asymptomatic, 1.32 fold higher in acute and 1.51 fold higher in comparison to normal samples (S1 Table). The comparative serum analysis of LF samples indicated that there could be significant alterations in the proteomics profile of LF patients following bancroftian infections.

## Zymography of serum matrix metalloproteinases

In order to identify novel LF biomarkers, we measured the serum levels of circulating gelatinases by gelatin zymography. The gelatinolytic activity of each MMP corresponds to a clear band in blue background representing the degradation of gelatin by serum gelatinases. In this study, 3 clear bands were visible in the serum samples of LF patients and healthy normal (Fig 3). The clear bands indicating proteolysis were localized at 240 kDa, 92 kDa and 72 kDa. The bands at 92 kDa and 72 kDa correspond to Gelatinase B (MMP-9) and Gelatinase A

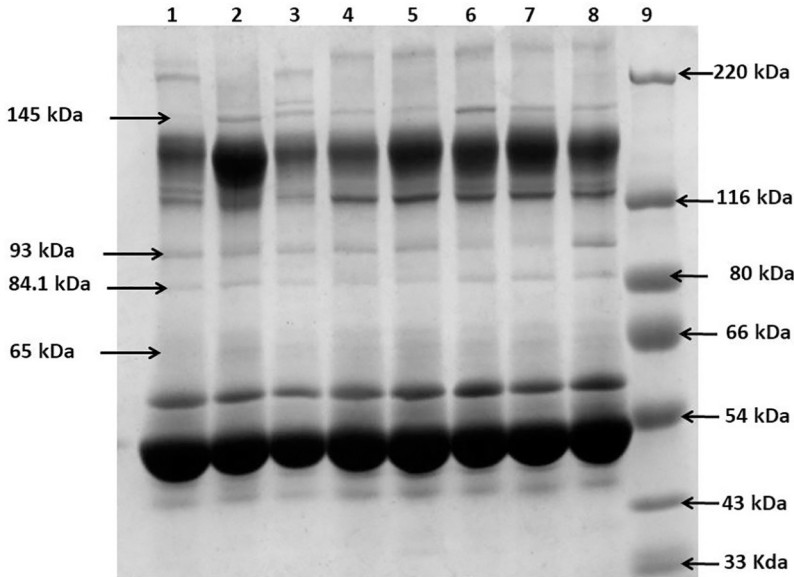

**Fig 2. 1 DE SDS-PAGE analysis of the normal and LF infected human sera.** Coomassie stained 7.5% SDS-PAGE of human serum sample, Lane 1. Normal-I, 2. Normal-II, 3. Asymptomatic-I, 4. Asymptomatic-II, 5. Acute-I, 6. Acute-II, 7. Chronic-I, 8. Chronic-II, 9. Mol. Marker.

(MMP-2) and the 240 kDa band is generally due to formation of MMP-9 dimers [27]. The proteolytic zone in each gel was subjected to densitometric analysis and normalized against a control serum. The highest molecular mass gelatinase of 240 kDa was 1.42, 1.83 and 2.59 fold more intense in asymptomatic, acute and chronic samples as compared to normal. The middle band of 92 kDa was 1.51 fold higher in asymptomatic, 1.94 fold higher in acute whereas 1.78 fold higher in chronic samples in comparison to normal samples. The lowest band at 72 kDa was 1.21 fold higher in asymptomatic, 1.31 fold higher in acute whereas 1.42 fold higher in chronic serums as compared with normal (S2 Table).

## 2DE analysis of serum proteomes following lymphatic filarial infections

We have also performed 2D electrophoresis for serum proteome analysis of LF-infected samples. The analysis was performed on 34 samples out of which 8 were acute, 8 were asymptomatic and 8 were chronic (Control samples = 10 normal). A representative 2D gel image of all LF categories and healthy normal is shown in Fig 4. Each LF clinical category gel image was compared with gel image of normal sample and analysis of differentially expressed proteins was done using PDQuest software. The gel image analysis revealed 161 spots in normal, 168 spots in asymptomatic, 159 spots in acute and 170 spots in chronic samples (S3 Table). A heat map was constructed to compare the differentially expressed proteins of LF samples (S3 Fig). On the basis of fold change calculations, out of 19 differentially expressed proteins (DEP) of LF samples, 13 were statistically significant ($p \leq 0.05$) and 6 DEP were highly significant ($p \leq 0.001$) (Table 3). The expression of down-regulated protein spots was in the range of -1.52 to -15.21 fold, while expression of up-regulated protein spots varied from +1.51 to +8.21 fold. The differentially expressed spots of LF samples that were statistically significant ($P < 0.05$) were subjected for further MALDI-TOF/MS and functional enrichment analysis.

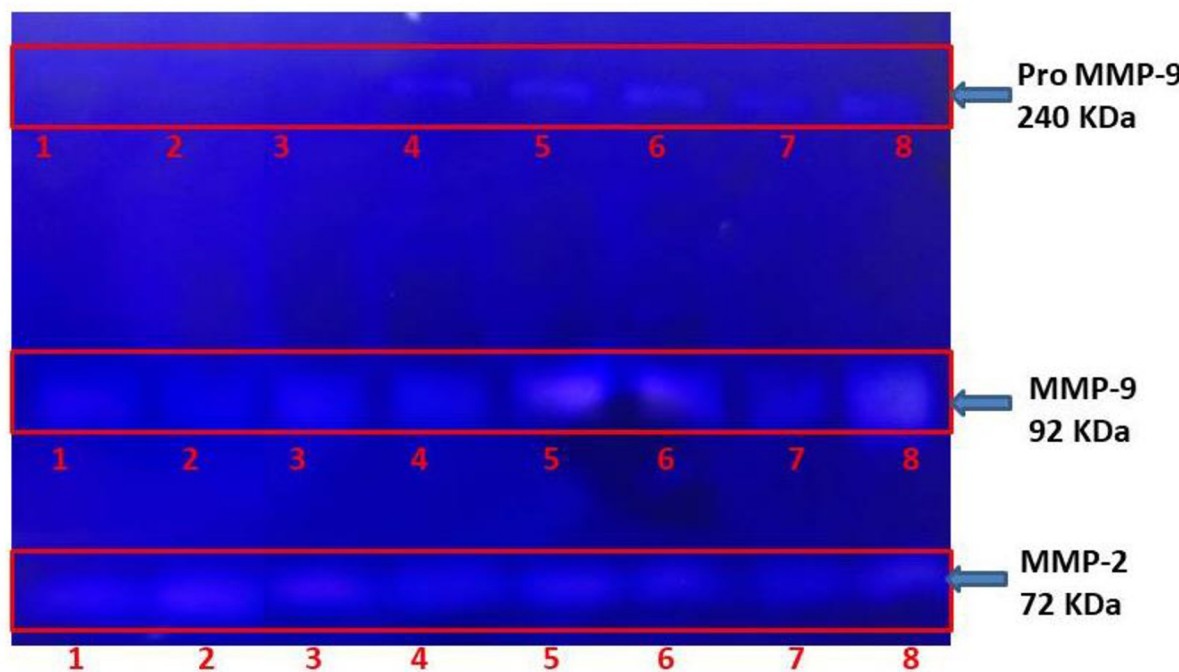

**A.**

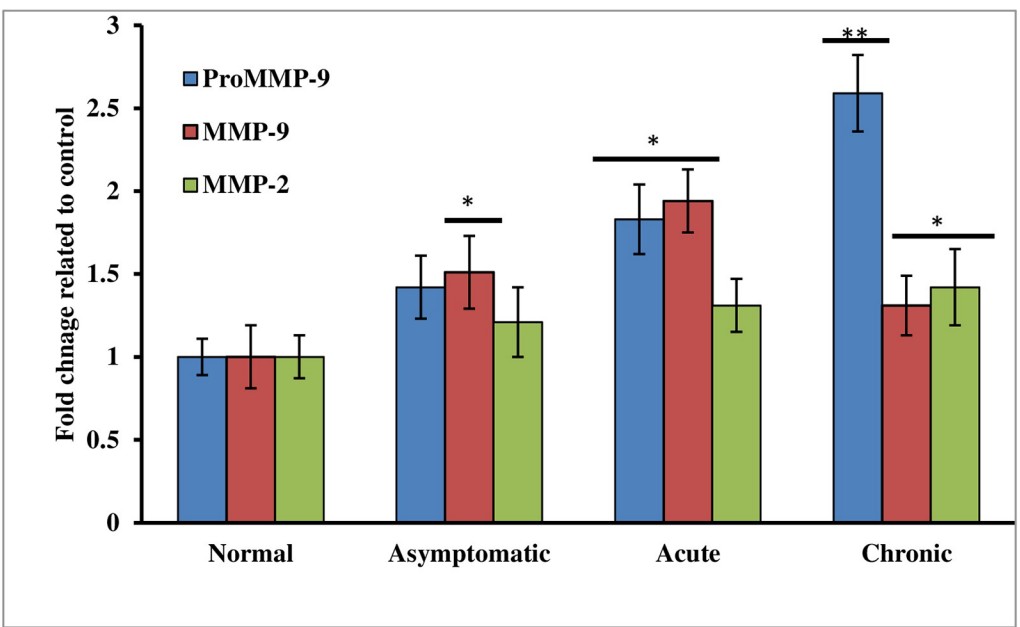

**B.**

**Fig 3. Gelatin zymography of normal and LF infected human sera.** A. A representative zymogram showing Gelatinase (MMP-2 and MMP-9) activity of normal and LF infected human sera (Lane 1. Normal-I, 2. Normal-II, 3. Asymptomatic-I, 4. Asymptomatic-II, 5. Acute-I, 6. Acute-II, 7. Chronic-I, 8. Chronic-II). B. A graph showing quantification of ProMMP-9, MMP-9, and MMP-2 by densitometry analysis of zymogram.

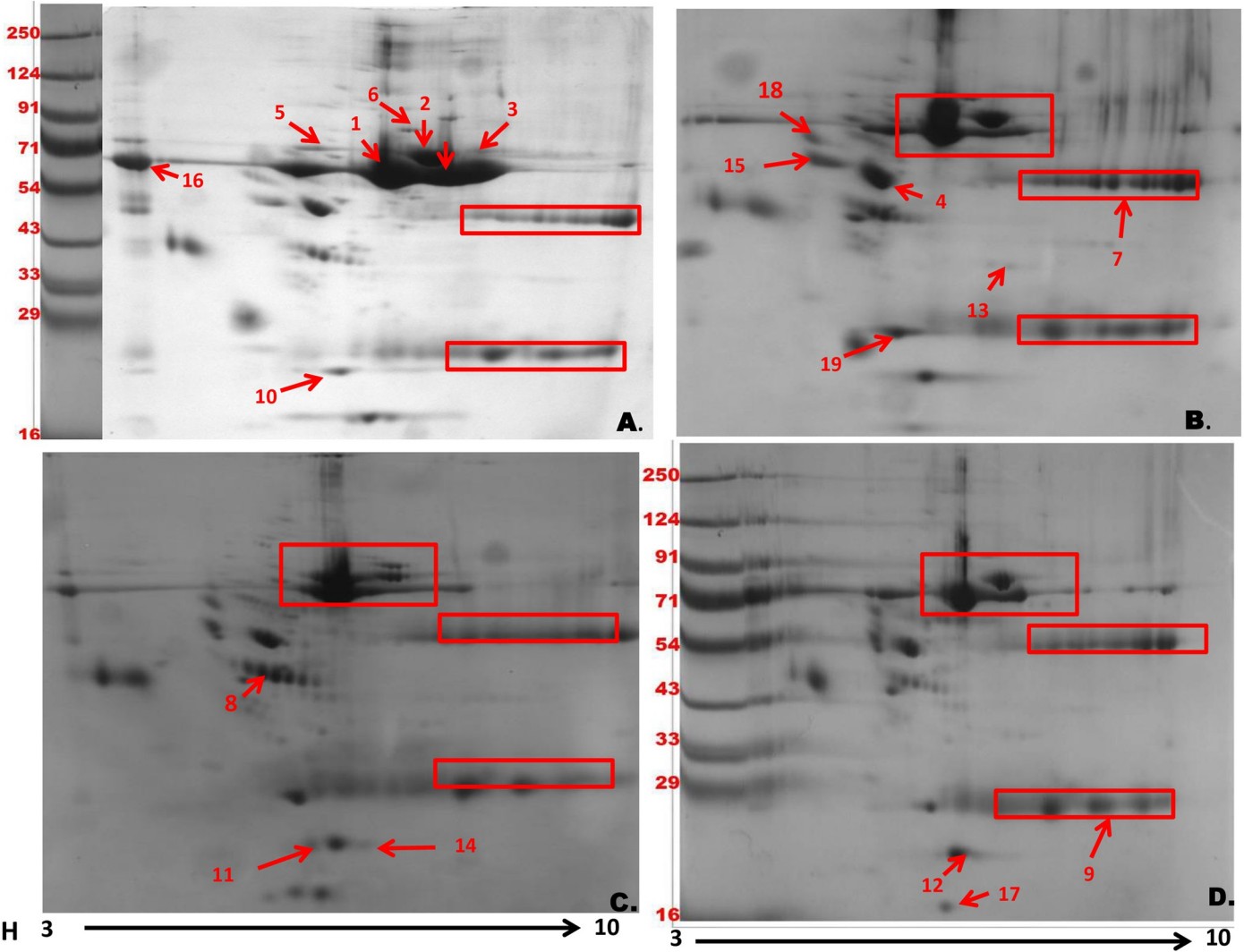

**Fig 4. Differential expression of human serum proteins in normal and LF infected patients identified in 2DE analysis.** Representative 2D gels of serum proteome form (A) Normal (n = 10), (B) Asymptomatic (n = 8), (C) Acute (n = 8) and (D) Chronic (n = 8) containing 400 µg depleted serum proteins. Serum protein samples focused on linear pH 3–10 IPG strips and separated on 10% SDS-PAGE gels, were stained with colloidal Coomassie stain. Out of 19 deferentially expressed spots 13 spots are statistically significant (P≤ 0.05) and 6 spots were highly significant (P≤ 0.01).

## Identification of differentially expressed proteins using MALDI-TOF/MS

Alterations of serum proteome due to the Lymphatic filarial infection resulted in differential expression of proteins as identified by SDS-PAGE and classical 2DE. The 4 DEP of SDS-PAGE were excised from the gel and subjected to MALDI-TOF/MS analysis to establish protein identity. The analysis revealed the identity of bands as Prothrombin (145 kDa), Haptoglobin (93 kDa), Alpha-1 antitrypsin (84.1 kDa) and kappa light chain (65 kDa). The 19 differentially expressed spots in classical 2DE were also identified by MALDI-TOF/MS and the result is summarized in Table 4. All the 4 DEP identified in 1D SDS PAGE, Prothrombin, Haptoglobin, Alpha-1 antitrypsin, and kappa light chain were also identified following 2D PAGE. The differential expression of proteins Albumin, Serotransferin, Complement C3 Precursor, Alpha-1 antitrypsin, Prothrombin, C-reactive protein on the basis of fold change was found to be highly significant. Serum amyloid A (SAA), was up-regulated in acute (1.91) and chronic

**Table 3. List of significant altered protein spots in the serum of normal and LF cases using 2DE image analysis done by PDQuest software.**

| S.N. | Spot Number | MW (kDa) | P Value (Anova test) | Fold Change Asymptomatic / Normal | Fold Change Acute / Normal | Fold Change Chronic/ Normal |
|------|-------------|----------|----------------------|-----------------------------------|----------------------------|-----------------------------|
| 1. | 1. | 67.4 | 0.014 | -2.2 | -2.8 | -5.3 |
| 2. | 2. | 73.2 | 0.0012 | -2.1 | -10.2 | -6.1 |
| 3. | 3. | 81.1 | 0.034 | -1.81 | -1.52 | -1.82 |
| 4. | 4. | 48.3 | 0.013 | +3.92 | +1.72 | +1.62 |
| 5. | 5. | 75.1 | 0.032 | -1.81 | -1.93 | -1.59 |
| 6. | 6. | 89.0 | 0.021 | -1.89 | -1.97 | -2.02 |
| 7. | 7. | 55.0 | 0.027 | +1.60 | +1.74 | +1.87 |
| 8. | 8. | 44.0 | 0.014 | +1.65 | +1.90 | -1.99 |
| 9. | 9. | 28.1 | 0.028 | +1.91 | +1.71 | +1.80 |
| 10. | 10. | 27.8 | 0.0042 | -10.2 | -11.1 | -15.21 |
| 11. | 11. | 17.3 | 0.031 | +1.91 | +2.21 | +1.62 |
| 12. | 12. | 19.9 | 0.021 | +2.21 | +1.91 | +2.01 |
| 13. | 13. | 31.0 | 0.0091 | +8.21 | +1.34 | +1.09 |
| 14. | 14. | 19.7 | 0.041 | -2.61 | -1.90 | -1.72 |
| 15. | 15. | 54.0 | 0.037 | +3.06 | +1.94 | +1.04 |
| 16. | 16. | 68.2 | 0.0022 | -4.29 | -3.92 | -2.92 |
| 17. | 17. | 17.1 | 0.013 | +1.05 | +1.91 | +2.32 |
| 18. | 18. | 56.3 | 0.052 | +2.12 | +1.81 | +1.51 |
| 19. | 19. | 28.0 | 0.0013 | +8.12 | +6.21 | +2.12 |

Plus sign indicates up-regulated whereas minus sign indicate down-regulated proteins in comparison to normal subjects.

**Table 4. MALDI-TOF/ MS analysis of differentially expressed serum protein normal and LF cases.**

| S.N. | Spot Number | Protein Name | Uniprot Accession Number | Sequence Length | Gene Name | Protein Score |
|------|-------------|--------------|--------------------------|-----------------|-----------|---------------|
| 1. | 1. | Albumin | P02768 | 609 | ALB | 211 |
| 2. | 2. | Serotransferin | P02787 | 698 | TF | 190 |
| 3. | 3. | Complement C3 Precursor | P01024 | 1663 | C3 | 179 |
| 4. | 4. | Alpha-1 antitrypsin | P01009 | 418 | SERPINAA1 | 123 |
| 5. | 5. | Prothrombin | P00734 | 622 | F2 | 89 |
| 6. | 6. | Complement Factor B | P00751 | 764 | CFB | 113 |
| 7. | 7. | Ig heavy chain gamma | P01859 | 326 | IGHG2 | 95 |
| 8. | 8. | Haptoglobin | P00738 | 406 | HP | 201 |
| 9. | 9. | Ig alpha-1 chain C region | P01876 | 353 | IGHA1 | 59 |
| 10. | 10. | Apolipoprotein A-I | P02647 | 267 | APOA1 | 176 |
| 11. | 11. | Ig kappa chain C region | P01834 | 116 | IGKC | 51 |
| 12. | 12. | Transthyretin | P02766 | 147 | TTR, PALB | 93 |
| 13. | 13. | Heterogenous nuclear riboprotein D like | 014979 | 420 | HNRNPDL | 87 |
| 14. | 14. | Sorcin | P30626 | 198 | SRI | 57 |
| 15. | 15. | Alpha-1B glycoprotein precursor | P04217 | 495 | A1BG | 163 |
| 16. | 16. | ER membrane protein complex subunit 10 | Q5UCC4 | 262 | EMC10 | 82 |
| 17. | 17. | Serum amyloid A | P35542 | 130 | SAA4, CSAA | 158 |
| 18. | 18. | Apolipoprotein A-IV | P06727 | 396 | APOP4 | 104 |
| 19. | 19. | C-reactive protein | P02741 | 224 | CRP | 91 |

Significant differentially expressed proteins in classical 2DE were identified by MALDI-TOF/MS.

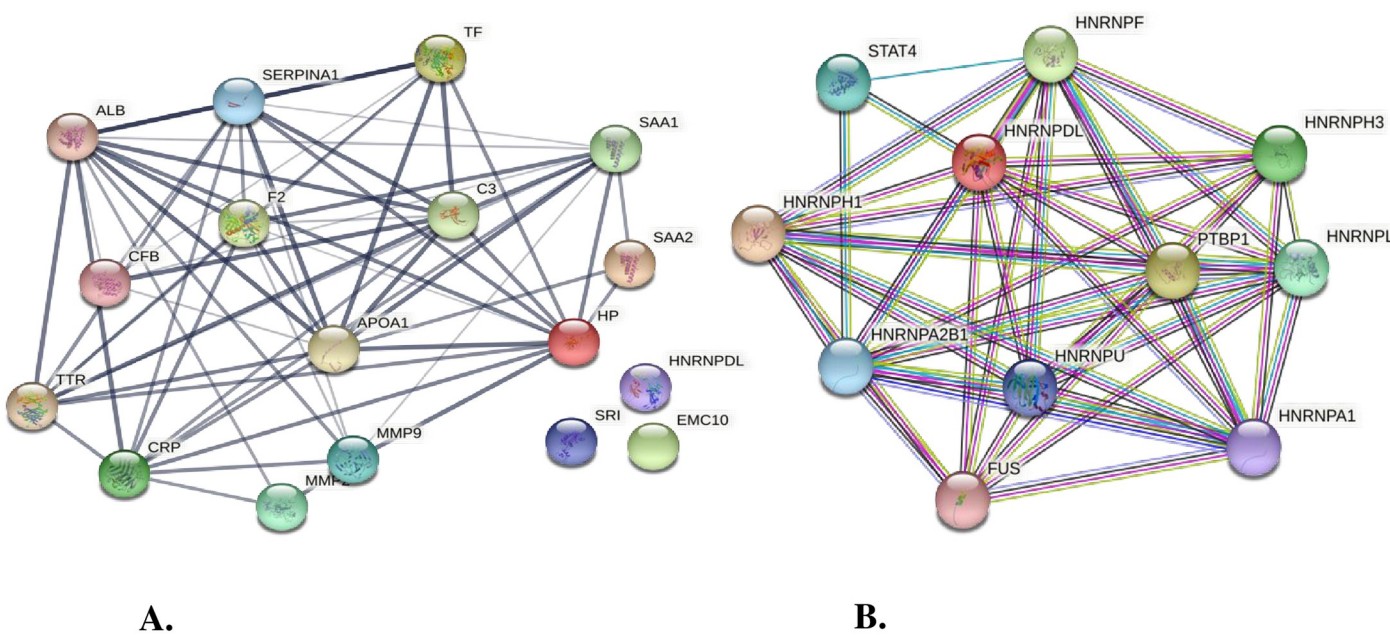

**Fig 5. STRING interaction network depicts the association among the differentially regulated proteins.** Color balls are representing individual proteins and lines represent their interactions: (A) Interaction of differentially expressed proteins and (B) hnRNP-D interactions with different proteins.

(2.32) cases although the expression was close to normal in asymptomatic samples. Serum Albumin was down-regulated by 2.2 folds in asymptomatic, 2.8 folds in acute and 5.3 folds in chronic samples when compared with normal serum. Apolipoprotein expression level was down-regulated by 10.2 fold, 11.1 fold and 15.21 fold in asymptomatic, acute and chronic cases. The study has also identified heterogeneous nuclear riboprotein D-like protein as a unique protein highly overexpressed in asymptomatic cases only.

## Protein networks and functional analysis

Protein-protein interaction network analysis was used in order to predict the functional interaction of differentially expressed proteins in LF cases. The STRING tool was employed and the interaction format was specified to "*Homo sapiens*" only (S4–S6 Tables). Out of 19 DEP, 14 proteins were inter-related to each other, whereas 3 proteins were not connected to any nodes and 2 were not found in STRING (Fig 5). Functional pathway analysis of 19 DEP of LF patients identified by MALDI-TOF/MS was investigated using DAVID & PANTHER softwares. The DEPs identified in this study were involved in 12 biological processes including proteolysis (16.98%), platelet degranulation (11.32%), acute phase response (11.32%), receptor mediated endocytosis (9.43%) and complement activation (9.43%). Concerning the cellular components, 12 component proteins were identified, which belonged to extracellular region (21.28%), extracellular exosomes (21.28%) and extracellular space (20.21%). The 12 molecular function obtained for the DEP included protein binding (34.78%), serine-type endopeptidase activity (19.57%) and identical protein binding (13.04%). In total, 4 KEGG pathways were obtained and the two major were complement and coagulation cascades (38.46%). The protein classes identified were defense/ immunity protein (27.78%), transferrin/carrier protein (22.22%) and protein modifying class (22.22%) (Fig 6).

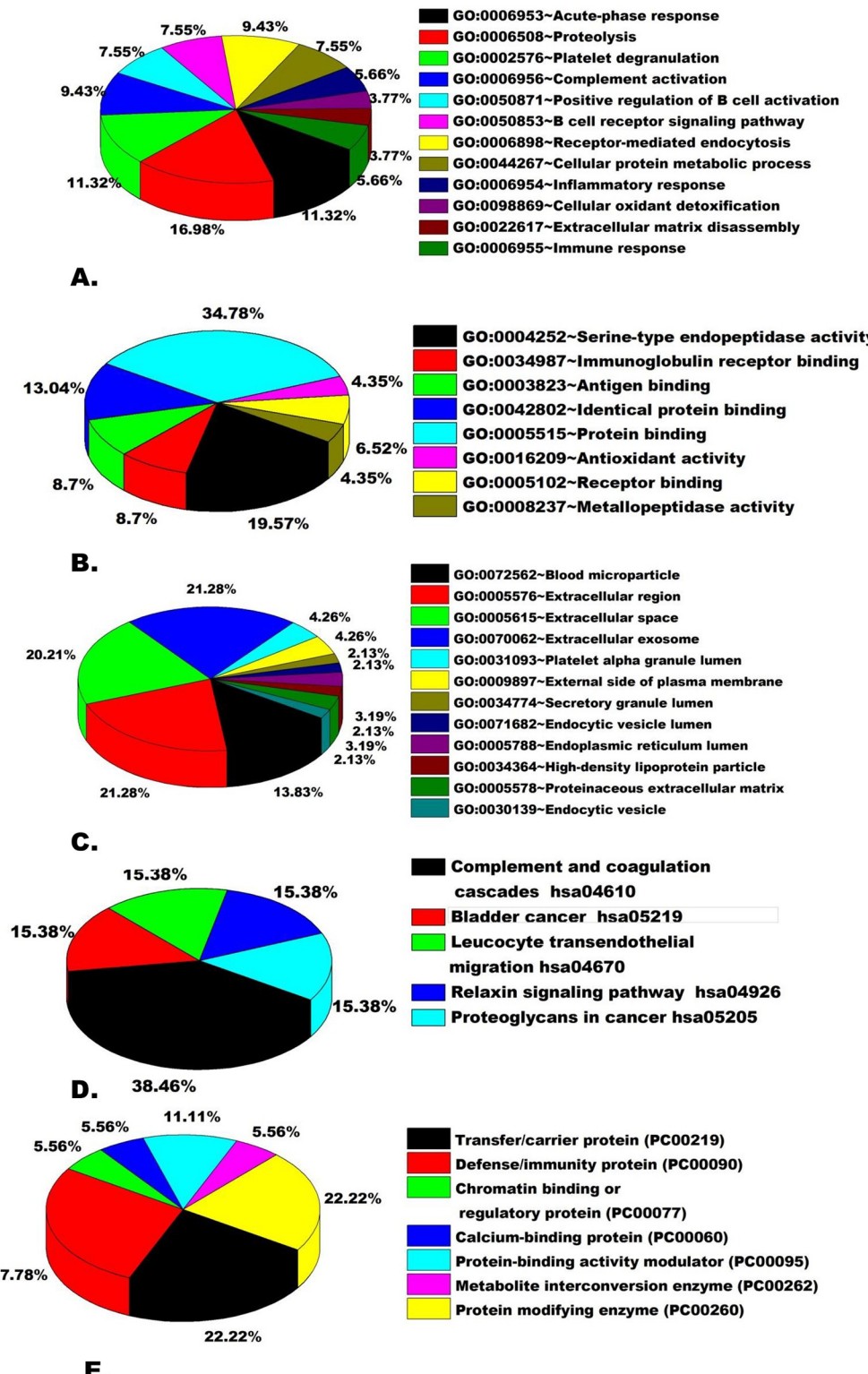

**Fig 6. Pie chart depicting the functional classification of differentially expressed proteins in LF patients.**
Differentially expressed proteins were classified: (A) Biological processes, (B) Molecular functions, (C) Cellular
components, (D) KEGG Pathway and (E) Protein class.

## Discussion

Around 863 million people in 47 countries of the world are still at risk of LF and require annual chemotherapeutic interventions to stop the further spread of LF infections. The success of the Global Program to Eliminate Lymphatic Filariasis largely depends upon the accurate assessment of LF transmission rates and intervention strategies. Lately, WHO's Diagnostic Technical Advisory Group (DTAG) for LF has accepted the need to identify new biomarkers that would be used for precision in GPELF implementation. The program decision-making will be largely boosted if sensitive biomarkers are available to distinguish past infections from active infections [28]. Several biomarkers that aid in disease diagnosis and are sensitive to disease progression have been identified by serum profiling. Human serum is a complex mixture of proteins, lipids, electrolytes, antigens hormones and other exogenous substances whose composition tends to swerve under disease conditions. In the present study, LF serum profiling was undertaken by performing FT-IR, MMP zymography, proteomics and Bio-informatics based analysis to identify new LF biomarkers which can be used for LF diagnosis and for differentiating among the different clinical stages of the disease.

In the present study, intensities of FT-IR peaks changed significantly during the progression of LF from one stage to another. Also, the FT-IR analysis revealed significant differences in peak intensities among different categories of LF serum samples which can be used for discriminating among different LF clinical stages. The FTIR peaks of Acute, Asymptomatic and Chronic LF samples was compared with control group for Fold change and Principle component analysis. The peaks at 1645 cm$^{-1}$, 1540 cm$^{-1}$ and 3300 cm$^{-1}$ were highest in asymptomatic cases so these peaks can be explored further for diagnosing early filarial infections. The peak at 1448 cm$^{-1}$ was of higher intensity in both asymptomatic and acute LF cases and thus could be investigated as a biomarker to distinguish past infections from active infections (S1 Fig). The routine LF detection is based on microscopic examination of thick blood smear for detection of microfilariae which essentially requires blood collection at odd hours in the night and well trained technicians. FT-IR is label free, inexpensive, and nondestructive spectroscopic technique, which can detect biochemical changes in serum samples. Lately, FT-IR is being implicated in the diagnosis of several infectious diseases such as malaria, zika virus infections, chikungunya and dengue [29, 30]. Consequently, FT-IR based LF diagnosis would be much faster and more economical, along with eliminating the need for cumbersome night blood collections.

In the present study, MMP-9 levels were significantly higher in all stages of LF patients as compared to normal and this overexpression was directly related to the disease severity. Lymphatic filariasis is often associated with changes in the architecture of the extracellular matrix (ECM) followed by superfluous accumulation of ECM components and matrix remodeling. The degradation and reorganization of ECM involve different types of matrix metalloproteinase (MMPs). ECM components elastin and type IV collagen are substrates for both MMP-2 and MMP-9 whereas MMP-2 can also break down interstitial collagen types I, II and III. MMPs are under stringent transcription and translational regulation besides being regulated by Tissue Inhibitors of Metalloproteinases (TIMPs) also. Earlier Anuradha et al., (2012) [31] have reported an increase in the levels of circulating MMPs and an imbalance between MMP/ TIMP ratios in filarial lymphedema patients by Luminex ELISA. Nevertheless, zymographic analysis of serum MMPs for establishing MMP-2/MMP-9 or other MMPs as LF biomarkers can be further elaborated by increasing the sample size.

Parasitic infections lead to rapid changes in expression patterns of circulating acute phase proteins and various pro-inflammatory cytokines [32], serum proteins, which can be directly correlated with disease progression. In our study, expressions of serum amyloid A (SAA),

alpha-1 antitrypsin, C—reactive protein (CRP), apolipoprotein A-IV and haptoglobin were significantly increased in LF samples as compared to normal samples. All the aforementioned proteins are Acute Phase Proteins (APPs) responsible for disease-related complications and pathogenesis [33]. Acute-phase proteins (APPs) are synthesized and released in response to infection, inflammation, or trauma. The APPs are involved in complement activation, in neutralizing enzymes, scavenging of free radicals and are also important in the host's immune response. CRP, serum amyloid P and serum amyloid A are the three most important APPs [34]. SAA can bind to microbial surfaces thus enhancing TNFα and IL-10 production from macrophages and respiratory burst of neutrophils. In addition, SAA is known for the induction of matrix metalloproteinases and other neutrophil-activating responses [35]. Higher blood concentrations of SAA are often associated with inflammation, infection, and tissue injury and in LF the expression of SAA increased with disease progression which can be correlated with the increase in inflammation in acute and chronic stages of the disease.

CRP, another upregulated APP identified in LF samples, is important for apoptosis, nitric oxide release, production of cytokines TNF-α & IL-6, and also for activating the classical complement pathway. In this study, CRP was overexpressed in all the LF samples when compared with normal serum. Interestingly, the CRP expression decreased with LF progression although the expression in chronic cases was almost twice that of normal samples. In normal individuals, the concentration of CRP is below 10 mg/L which can increase several folds following bacterial infection and in certain diseases like rheumatoid arthritis, cardiovascular complications, and progressive tumors [36], hence increase in CRP levels is mostly associated with diseases. An increase in CRP following LF infection has been reported earlier [37]. but our proteomics results confirm increase of CRP levels in microfilaremic and acute LF cases also.

APP alpha-1 antitrypsin (AAT) another APP also followed the same trend and the expression although normal samples had the lowest AAT. AAT is an anti-inflammatory and anti-infective molecule functioning in the regulation of the host's inflammatory responses. The increase in AAT and CRP following LF infection and the gradual decrease in their levels from asymptomatic stage to chronic could be attributed to the immune-modulating capabilities of filarial parasites.

The two-dimensional PAGE showed significant alterations in the levels of serum apolipoprotein A-I and A-IV. Serum apolipoprotein A-I, expression levels were almost 10 to 15 folds downregulated in LF samples and this decrease was directly related to the disease progression. ApoA-I has anti-inflammatory and anti-oxidant functions which would get highly compromised due to its significantly lower expression levels in LF patients. The expression of serum apolipoprotein A-IV was upregulated in LF serum samples as compared with normal. Apolipoprotein A-IV has been demonstrated as an early marker of kidney failure and the higher levels of this protein in LF samples can be a strong indication of progressive kidney damage [38]. LF patients had downregulated expression of serum Albumin which was in the range of -2.2 folds in asymptomatic to -5.3 folds in chronic cases in relation to normal samples. Lower levels of albumin are generally due to inflammation, liver or kidney diseases. LF patients often have underlying liver and kidney damages hence further studies are needed to understand the physiological reasons of hypoalbuminemia seen in LF.

We have observed significantly high levels of hnRNP-D (heterogeneous nuclear ribonucleoprotein-D) in asymptomatic samples (8.21 folds) which tapered back to normal in acute (1.34 folds) and chronic cases (1.09 folds). Functional network analysis has revealed the interaction of hnRNP-D with other hnRNPs like hnRNP-F, hnRNP-H3, hnRNP-L, hnRNP-A2B1, hnRNP-U, hnRNP-A1, and proteins Polypyrimidine tract- binding protein 1 (PTBP1), Signal transducer and activator of transcription 4 (STAT4), and RNA-binding protein FUS (FUS) protein. hnRNPs play an important role in the formation of mRNA from pre mRNAs/

hnRNAs. They are also involved in cellular transport and translation of mRNAs. Different hnRNPs have different functions depending upon their cellular localization. hnRNP-D mediates rapid decay of mRNA by binding with mRNA destabilizing sequences. Earlier hnRNP-D has been implicated in leishmaniasis infections also where its increased expression was observed following *Leishmania donovani* infection of human monocyte-derived macrophages [39]. Overexpression of hnRNP-D has been correlated with poor prognosis in oral squamous cell carcinoma patients and is being considered as a potential target for molecular therapeutics [40]. In our study, an increase in hnRNP can be speculated as a worm-induced proparasitic response; still, further elucidation of the molecular mechanism is essential for delineating its role in LF.

Another interesting observation was that complement precursor C3 and complement factor B were down-regulated in all categories of LF patients. On one hand, the complement C3 is essential for activation of the complement system and on the other hand cleavage products of complement C3 also act as opsonins. Complement factor B is a member of the alternative pathway and provides the catalytic activity of C3 convertase. The downregulation of both complement precursor C3 and complement factor B in serum samples of LF patients will greatly impact classical as well as alternative pathways of the complement cascade. The decrease in the expression of complement proteins in LF disease has not been reported till date, to the best of our knowledge. The present study was done on proteome profiles of different human volunteers and therefore, we observed heterogeneity in results. No doubt the biological variations of age, environment, diet, and gender will be crucial in the detection of LF biomarkers. Nevertheless, the markers that have been detected by FT-IR and proteomic analysis in the present report seem promising from the diagnostic point of view.

## Conclusion

To our knowledge, this is the first report of comparative human serum profiling in Lymphatic Filarial patients to identify *Wuchereria bancrofti* specific host biomarkers using classical 2DE gel analysis. The FT-IR analysis, MMP zymography and classical 2DE combined with MALDI-TOF/MS has unveiled several LF-specific biomarkers which certainly need to be explored further. The proteomic analysis in the study clearly showed that most of the altered proteins have crucial physiological functions. Additionally, some of the differentially expressed proteins are also essential for producing a balanced immune response. Hence, further studies on the biomarkers identified in this report are imperative to develop these as established diagnostic markers of LF infections and also for a better understanding of disease pathogenesis.

## Supporting information

**S1 Fig. Fold Chain analysis of Asymptomatic (A), Acute (B) and Chronic (C,) LF case with Healthy Normal, the figures correspond to FT-IR spectroscopy.**
(PDF)

**S2 Fig. PCA score plot of normal and LF patient serum by using FT-IR spectral data.** A) normal and asymptomatic, B) normal and acute and C) normal and chronic data.
(PDF)

**S3 Fig. Heat map for comparative analysis of different stages of LF cases after image analysis by PD-quest software.**
(PDF)

**S1 Table. List of differentially expressed protein bands in the serum of normal (control) and LF cases using SDS-PAGE, image analysis done by Quantity one and image J software.** (DOCX)

**S2 Table. List of differentially expressed protein spots in the serum of normal (control) and LF cases using gelatin zymography, image analysis done by Quantity one and image J software.** (DOCX)

**S3 Table. 2 D gel image comparative analysis by PDQUEST software for normal and LF cases.** (DOCX)

**S4 Table. Interacting proteins and their string ID.** (DOCX)

**S5 Table. Interacting proteins and their string ID and combining score.** (DOCX)

**S6 Table. Heterogeneous nuclear ribonucleoprotein-D like interaction with different partners.** (DOCX)

**S1 Raw images.** (PDF)

## Acknowledgments

We are thankful to all the volunteers who have participated in our study. AS and VK are grateful to SERB for providing research project (SR/FT/LS-177/2010). VK is grateful to the Department of Biotechnology (DBT), New Delhi for providing Senior Research Fellowship (DBT/JRF/BET-18/I/2018/AL/19). AM is thankful to the Council of Scientific and Industrial Research (CSIR), New Delhi for providing Senior Research Fellowship (09/013(0832)/2018-EMR-I). SR is acknowledged to UGC-BSR faculty fellowship (Ref. no. F.18-1/2011 (BSR) 26 June 2018). The authors are also grateful to ISLS BHU for providing 2D PAGE facility and laboratory space.

## Author Contributions

**Conceptualization:** Vipin Kumar, Awadehesh Kumar Yadav, Sushma Rathaur, Anchal Singh.

**Data curation:** Vipin Kumar, Anchal Singh.

**Formal analysis:** Vipin Kumar, Ayushi Mishra, Anchal Singh.

**Investigation:** Anchal Singh.

**Methodology:** Vipin Kumar, Ayushi Mishra.

**Resources:** Awadehesh Kumar Yadav, Sushma Rathaur, Anchal Singh.

**Supervision:** Anchal Singh.

**Validation:** Vipin Kumar, Ayushi Mishra, Anchal Singh.

**Visualization:** Vipin Kumar, Ayushi Mishra.

**Writing – original draft:** Vipin Kumar, Anchal Singh.

**Writing – review & editing:** Vipin Kumar, Ayushi Mishra, Anchal Singh.

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
