## [Decision Letter · Decision Letter 0]

31 Mar 2022

PONE-D-22-04328Lymphatic Filarial Serum Proteome Profiling for Identification and Characterization of Diagnostic BiomarkersPLOS ONE

Dear Dr. Singh,

Thank you for submitting your manuscript to PLOS ONE. After careful consideration, we feel that it has merit but does not fully meet PLOS ONE’s publication criteria as it currently stands. Therefore, we invite you to submit a revised version of the manuscript that addresses the points raised during the review process.

We look forward to receiving your revised manuscript.

Kind regards,

Suprabhat Mukherjee, Ph.D.

Academic Editor

PLOS ONE

Journal Requirements:

Additional Editor Comments:

Authors need to revise the manuscript. Alongside the comments of the reviewers, please modify the discussion and include important findings (e.g. https://doi.org/10.1093/infdis/jix067; 10.1038/s42003-019-0392-8) to highlight the importance of the present study.

Reviewers' comments:

Reviewer's Responses to Questions

**Comments to the Author**

1. Is the manuscript technically sound, and do the data support the conclusions?

Reviewer #1: Partly

Reviewer #2: Partly

2. Has the statistical analysis been performed appropriately and rigorously? 

Reviewer #1: No

Reviewer #2: No

3. Have the authors made all data underlying the findings in their manuscript fully available?

Reviewer #1: Yes

Reviewer #2: No

4. Is the manuscript presented in an intelligible fashion and written in standard English?

Reviewer #1: Yes

Reviewer #2: Yes

5. Review Comments to the Author

Reviewer #1: I have read with interest the manuscript “Lymphatic Filarial Serum Proteome Profiling for Identification and Characterization of Diagnostic Biomarkers” having MS no: PONE-D-22-04328

In the current study, the authors collected the serum of persons living in endemic areas of LF and analyzed the presence and abundance of C-reactive protein, α-1-antitrypsin, heterogeneous nuclear ribonucleoprotein D like apolipoproteins A-I and A-IV. The authors projected them as biomarkers of LF and proposed them as diagnostic measures. This work sounds interesting and does have fair medical importance, however, I have some concerns over the manuscript that should be addressed properly.

The major problems with this manuscript are:

1. Correctly denote the full name of GPELF where used, it has been denoted differently in places, authors should have been more careful.

2. Authors mentioned that “The proteomics analysis results showed that various proteins were differentially expressed (p<0.05), including C-reactive protein, α-1-antitrypsin, heterogeneous nuclear ribonucleoprotein D like, apolipoproteins A-I and A-IV which have not been reported in Lymphatic Filariasis previously”; However, according to available literature the claim appeared far more enthusiastic and not based on facts as there are reports. Some of them are

CRP:

i. J Clin Immunol. 1991 Jan;11(1):46-53. doi: 10.1007/BF00918794.

ii. https://doi.org/10.1371/journal.ppat.1002749

α-1-antitrypsin:

i. Indian J Med Res. 2011 Jul; 134(1): 79–82.

ii. Human Parasitic Pulmonary Infections; Gary W. Procop, Ronald C. Neafie, in Pulmonary Pathology (Second Edition), 2018; Tropical Pulmonary Eosinophilia and Aberrant Filaria

Apolipoproteins:

i. https://www.thelancet.com/pdfs/journals/lancet/PIIS0140-6736(06)69100-9.pdf

ii. doi: 10.1002/14651858.CD003753.pub4

The claim should be justified properly and I suggest a thorough revision of the literature of the manuscript literature and citations they had made.

3. The authors claimed that “To our knowledge, this is the first report of comparative human serum profiling in different categories of LF patients.” This is partly true as there are reports on IgG4 antibodies on IgE-activated granulocytes in patients. (doi: 10.1371/journal.pntd.0005777).

I do request the authors a further detailed survey of existing literature before confirming any claim. Though the merit of a research article may not decrease the unnecessary and false claim can lead the readers to a fix and discourage them from a further study on the particular aspect.

4. Regarding the supporting information, I suggest that the authors should upload a single pdf or Docx supplementary file to ease the reading of the reviewer and readers.

5. Serum profiling is alright. However, what they have claimed that is a maiden report is not there are reports of the serum profiling and precisely for some of the proteins they emphasize on. How these serum or precise combinations of them can be of diagnostic purpose should be elaborated and given more emphasis. Additionally why anyone would opt for these complex and costly methods against the available methods is not properly elaborated. Are they are thinking of any on-field diagnosis? Then this should also be elaborated.

6. In the introduction, the epidemiological status should be cited properly.

7. In the introduction, section Authors should elaborate in a few sentences what are the available immunoassay tests and their diagnostic criteria, restrictions, limitations, and scope of further and other immunoassay diagnostics. There are a few test kits already available and authors choose to completely avoid them.

8. In the Materials and Method section and to the heading Clinical characterization of LF cases: "Lymphatic Filariasis.......secondary infections.” this should go to the introduction. There is no page or line number, please provide the page and line number to your manuscript.

9. What are the selection criteria of the participants? Are they were previously screened for LF? What is their age, sex? if they are selected randomly based on their consent (it should be properly mentioned) then the percentage of population positive for LF (77%) is very hard to believe. and this should be properly verified and revised.

10. Separation of immune complexes: By the method of whom?

11. In the Result section: Authors denoted that "serum protein concentration altered

Significantly in response to LF infections", however, the applied method of significance is not mentioned and if there is any significant change between groups of infected peoples, i.e. asymptomatic, acute and chronic is not properly mentioned. Authors should employ extensive statistical means of exploring serum parameters between these groups and with the non-infected ones when advocating these parameters for diagnostic means.

12. As the authors just predicted the functional interaction the heading should be written like that. For a proper evaluation of functional interaction, a far more detailed study is needed with inhibitors and siRNA and other detailed molecular biology methods.

Reviewer #2: The present study evaluates the proteome profile of serum from individuals infected with Lymphatic Filariasis as a possible source for identification and characterization of diagnostic biomarkers that may be useful in detecting LF infections in asymptomatic cases and can also serve as indicators for differentiating among different clinical stages of the disease. Based on the results obtained, the authors attest to the potential of human serum profiling for detection of LF and claim that their study is the first report of comparative human serum profiling in different categories of LF patients.

The study addresses an important area of identifying LF biomarkers in active infections undergoing IDA so as to assess the outcome of GPELF, however, substantial concerns remain that needs to be addressed before the work is accepted for publication.

Reviewer’s comments:

The study dwells on an important area of human health that is of major concern in many developing countries. However, the study even though concise, suffers from major drawbacks that limits the publication of this piece of work in the present form. The authors need to address the following concerns.

Major concerns:

1. LF patients were infected with W. bancrofti. But the authors have not segregated them on the basis of age and gender. This should have been done and data presented accordingly. Also, it would have been interesting if some B. malayi infected cases were also part of the study as it would have demonstrated the robustness of the methodologies used. Nevertheless, the data presented should be segregated and discussed keeping in mind the differences between gender and age.

2. The authors claim that serum protein concentration was significantly altered in response to LF infections (Results section-1st Paragraph). However, the values of protein concentration (Table-01) in normal serum (69 ± 8.0 mg/ml), asymptomatic (78 ± 12.0 mg/ml), acute cases (81 ± 14 mg/ml) and chronic cases (93 ± 19 mg/ml) do NOT look drastically different, especially when one looks at the SD value carefully between groups. The authors have indicated p values at three different places in table 1, but it is very very difficult to comprehend how come with such high SD value between groups, the authors could still generate p value < 0.05. The authors must work out the statistical data again or clearly mention which statistical tool was used to calculate the p value. Same concern holds true for IFF and immune complex data as well.

3. Fold change analysis of FT-IR data shows significantly altered peaks at 3300, 2950, 1645, 1540 and 1448 cm-1 between asymptomatic and acute cases. What about the altered peaks in chronic cases? The authors need to discuss this point.

4. The authors claim that a combination of different techniques (FTIR, MMP zymography, SDS-PAGE, 2DGE, MALDI-TOF/MS etc.) was used to identify LF biomarkers from serum samples of different stages of LF patients. How do the authors conclude that the biomarkers thus detected were due to LF infection only and not due to any other previous ailment or other diseased conditions? Merely writing, “LF infected cases were examined by a clinician and were categorized based on the above mentioned manifestations and presence/absence of microfilariae in the bloodstream” does not hold much value as patients exhibiting Tropical Pulmonary Eosinophilia do not show presence of Mf in the peripheral blood. What was the criteria used for exclusion by the clinician, needs to be mentioned in detail.

5. Can the adopted approach be used to distinguish LF infection caused by Wuchereria bancrofti, Brugia malayi and Brugia timori? This needs to be discussed by the authors.

6. What was the duration of the study? Since acute infection cases would progress to chronic cases, won’t their serum profile differ over a period of time and match with the chronic cases? What were the time points of blood withdrawal? Single time point withdrawal is not sufficient to provide the complete picture as gravid females would continuously produce Mf which may alter serum profile? Also, age and gender would make a difference.

7. Zymography and MMP data is not very convincing, as very small differences exist between groups. When one takes SD into account, there would be hardly any differences.

8. Fig 2 , Marker well has different contrast (top and bottom) how come?

Minor concerns:

1. The authors need to factually correct the starting line of abstract and introduction. As per latest WHO figures, 863 million people are affected by LF, the number citing 120 million infections is a very old data (https://www.who.int/news-room/fact-sheets/detail/lymphatic-filariasis).

2. Under “Separation of immune complexes”, either the authors must mention the name of author/research group or write “as described previously”. Just writing “LF cases were isolated by the method of [15] with minor modification” is not correct.

3. Length of IPG strips and pH gradient used in 2DGE should be included in M & M, rather than just in the results.

4. Fig 3b, y axis should be “change”

5. Clearly state statistical comparisons have been made between which groups in fig 1 a and 1b?

6. In Abbreviation and abstract, GPELF should read as Global Programme to Eliminate Lymphatic Filariasis.

6. PLOS authors have the option to publish the peer review history of their article (what does this mean?). If published, this will include your full peer review and any attached files.

Reviewer #1: **Yes: **Niladri Mukherjee

Reviewer #2: No

---

## [Author Response · Author response to Decision Letter 0]

25 May 2022

Dear Editor, 

We are thankful to you and the reviewers to critically review our manuscript and consider it for publication in your esteemed journal. We sincerely appreciate the efforts of reviewers in the COVID pandemic for taking out time and giving us valuable suggestions and comments, for improving the quality of this research article. We would also like to update the data availability statement as we are submitting the raw gel/blot images as S1_raw_images in supporting information.

 Please find herewith the revised manuscript incorporating the corrections suggested by the reviewers. The response to the reviewer’s comments has been addressed in a point-by-point manner and highlighted in yellow in this text, Looking forward for a positive reply. 

Thanking you. 

Sincerely,

Dr. Anchal Singh

(Corresponding Author)

 

Response Letter

Comments to the Author 

1. Is the manuscript technically sound, and do the data support the conclusions?

Ans. We have taken control in each experiment and every experiment was performed in duplicate and triplicate. In our study 87 participants were selected from an endemic area for blood collection in which 23 were asymptomatic, 19 were acute, 25 were chronic patients and 20 were healthy individuals (Normal).

2. Has the statistical analysis been performed appropriately and rigorously?

Ans. Yes, we have performed statistical analysis in each table and experimental data The data is expressed as mean ± SD by using two tailed student’s t-test and one way Anova (Origin and Metaboanalyst software), which is now mentioned under materials and methods paragraph Statistical Analysis (line number from 272 to 277).

3. Have the authors made all data underlying the findings in their manuscript fully available?

Ans. Yes, fully available in supplementary information.

4. Is the manuscript presented in an intelligible fashion and written in Standard English?

Ans. Yes

5. Review Comments to the Author

Reviewer # 1 Comments

Comments 1 Correctly denote the full name of GPELF where used, it has been denoted differently in places, authors should have been more careful.

Ans. Now the full name of GPELF (Global Programme to Eliminate Lymphatic Filariasis) is correctly denoted and added in the manuscript.

Comments 2 Authors mentioned that “The proteomics analysis results showed that various proteins were differentially expressed (p < 0.05 ) including C-reactive protein, α-1-antitrypsin, heterogeneous nuclear ribonucleoprotein D like, apolipoproteins A-I and A-IV which have not been reported in Lymphatic Filariasis previously”; However, according to available literature the claim appeared far more enthusiastic and not based on facts as there are reports. Some of them are

CRP: i. J Clin Immunol. 1991 Jan;11(1):46-53. doi: 10.1007/BF00918794. 

ii. https://doi.org/10.1371/journal.ppat.1002749

α-1-antitrypsin: i. Indian J Med Res. 2011 Jul; 134(1): 79–82.

 ii. Human Parasitic Pulmonary Infections; Gary W. Procop, Ronald C. Neafie, in Pulmonary Pathology (Second Edition), 2018; Tropical Pulmonary Eosinophilia and Aberrant Filaria

Apolipoproteins: i. https://www.thelancet.com/pdfs/journals/lancet/PIIS0140-6736(06)69100-9.pdf ii. doi: 10.1002/14651858.CD003753.pub4 

The claim should be justified properly and I suggest a thorough revision of the literature of the manuscript literature and citations they had made

Ans As per suggestion of reviewer the manuscript is thoroughly revised and the relevant references are added in the manuscript (yellow in color) (line number from 444 to 445 and 467 to 469). Although these proteins have been analyzed earlier, we are investigating the LF clinical stage specific expression of various serum proteins, for the first time using proteomics.

Comment 3 The authors claimed that “To our knowledge, this is the first report of comparative human serum profiling in different categories of LF patients.” This is partly true as there are reports on IgG4 antibodies on IgE-activated granulocytes in patients. (doi: 10.1371/journal.pntd.0005777).

Ans We have modified the sentence, “To our knowledge, this is the first report of comparative serum profiling in different clinical stages of LF using classical 2DE gel analysis” (line number from 518 to 520 ). 

Comment 4. Regarding the supporting information, I suggest that the authors should upload a single pdf or Docx supplementary file to ease the reading of the reviewer and readers.

Ans We have uploaded a single PDF of supplementary file.

Comment 5. Serum profiling is alright. However, what they have claimed that is a maiden report is not there are reports of the serum profiling and precisely for some of the proteins they emphasize on. How these serum or precise combinations of them can be of diagnostic purpose should be elaborated and given more emphasis. Additionally why anyone would opt for these complex and costly methods against the available methods is not properly elaborated. Are they are thinking of any on-field diagnosis? Then this should also be elaborated

Ans The most common LF diagnostic methods are ICT (immune-chromatographic) Test, Lymphoscintinography, X-ray diagnosis and Ultrasound but these methods have limited sensitivity, high cost and cumbersome sample processing. The Asymptomatic patients have no external signs of LF infection and hence remain undiagnosed until chronic symptoms appear. The popular night blood film examination does not show positive result if microfilaremia load is low.

Although the serum comparative proteome analysis can be a bit expensive but the study can pave the way for developing potential biomarkers for LF infections. In the future, the laboratory is planning to select one or two biomarkers from the proteomic studies which could be used for identification of different stages of LF infection using more convenient, cost effective techniques like ELISA or FTIR (line number from 140 to 145 and 151 to 155).

Comment 6. In the introduction, the epidemiological status should be cited properly

Ans Cited properly according to journal and highlighted in yellow (line number 86).

Comment 7. In the introduction, section Authors should elaborate in a few sentences what are the available immunoassay tests and their diagnostic criteria, restrictions, limitations, and scope of further and other immunoassay diagnostics. There are a few test kits already available and authors choose to completely avoid them

Ans Yes, elaborated in manuscript introduction and highlighted in yellow (line number from 105 to 114).

Comment 8. In the Materials and Method section and to the heading Clinical characterization of LF cases: "Lymphatic Filariasis.......secondary infections.” this should go to the introduction. There is no page or line number, please provide the page and line number to your manuscript.

Ans, Now, we have made the changes and the quoted text is moved under Introduction (line number from 131 to 139). Page number and line number has been added.

Comment 9. What are the selection criteria of the participants? Are they were previously screened for LF? What is their age, sex? if they are selected randomly based on their consent (it should be properly mentioned) then the percentage of population positive for LF (77%) is very hard to believe. and this should be properly verified and revised.

Ans- We have collected samples from LF positive cases who had come for LF diagnosis in National Centre for Disease Control, Ministry of Health and Family Welfare, Varanasi. In total we collected samples from 87 LF cases and 20 healthy controls. For control samples the blood was collected from healthy volunteers living in same endemic area but were free from LF infections. The LF infection and clinical stage was confirmed by a clinician after which blood was collected from the patients.

Comment 10. Separation of immune complexes: By the method of whom?

Ans The immune complexes were separated by using method of Menikou et al., 2019 with minor modifications cited in the manuscript and highlighted in yellow (line number 194). 

Comment 11. In the Result section: Authors denoted that "serum protein concentration altered significantly in response to LF infections", however, the applied method of significance is not mentioned and if there is any significant change between groups of infected peoples, i.e. asymptomatic, acute and chronic is not properly mentioned. Authors should employ extensive statistical means of exploring serum parameters between these groups and with the non-infected ones when advocating these parameters for diagnostic means.

Ans: The significant serum protein alteration was seen only in chronic cases (P < 0.05) as compared to control samples. We have added this information in the manuscript in result section line number 280 to 289. P value was calculated by Origin 8.0. The P value (p <0.05) was considered as significant with respect to control. Now we have added a paragraph of statistical significance under materials and methods showing the statistical tests applied in each experiment (line number from 271 to 277).

 Comment 12 As the authors just predicted the functional interaction the heading should be written like that. For a proper evaluation of functional interaction, a far more detailed study is needed with inhibitors and siRNA and other detailed molecular biology methods.

Ans – Heading is now modified as Protein networks and functional analysis under materials and methods and results section.

Reviewer #2

Comment 1. LF patients were infected with W. bancrofti. But the authors have not segregated them on the basis of age and gender. This should have been done and data presented accordingly. Also, it would have been interesting if some B. malayi infected cases were also part of the study as it would have demonstrated the robustness of the methodologies used. Nevertheless, the data presented should be segregated and discussed keeping in mind the differences between gender and age.

Ans- This study was performed between age group 20 to 60 years LF cases. The selected area for this study is eastern zone of Uttar Pradesh, India which is the endemic only for W. bancrofti, in this zone B. malayi and B. timori infections are not found. In India B. malayi infection occur only in a narrow zone of southern India, hence, we have not included B. malayi in this study.

For this study we have classified LF cases according to standard WHO guidelines for clinical characterization into asymptomatic, acute and chronic cases. Earlier reports on LF have also used similar clinical stage wise classification only 

1. Prodjinotho UF, von Horn C, Debrah AY, et al. Pathological manifestations in lymphatic filariasis correlate with lack of inhibitory properties of IgG4 antibodies on IgE-activated granulocytes. PLoS Negl Trop Dis. 2017;11(7):e0005777. Published 2017 Jul 24. doi:10.1371/journal.pntd.0005777.

2. Anuradha R, George PJ, Pavan Kumar N, et al. Circulating microbial products and acute phase proteins as markers of pathogenesis in lymphatic filarial disease. PLoS Pathog. 2012;8(6):e1002749. doi:10.1371/journal.ppat.1002749.

Comment 2. The authors claim that serum protein concentration was significantly altered in response to LF infections (Results section1st Paragraph). However, the values of protein concentration (Table-01) in normal serum (69 ± 8.0 mg/ml), asymptomatic (78 ± 12.0 mg/ml), acute cases (81 ± 14 mg/ml) and chronic cases (93 ± 19 mg/ml) do NOT look drastically different, especially when one looks at the SD value carefully between groups. The authors have indicated p values at three different places in table 1, but it is very very difficult to comprehend how come with such high SD value between groups, the authors could still generate p value < 0.05. The authors must work out the statistical data again or clearly mention which statistical tool was used to calculate the p value. Same concern holds true for IFF and immune complex data as well.

Ans- The serum protein concentration were elevated in response to LF infection, as disease progresses from one stage to another the protein concentration increased. The data were expressed as mean ± SD by using two tailed Student’s t-test and P-value is provided in the manuscript, the significant serum protein alteration was seen only in chronic cases (P < 0.05) as compared to control samples. We have added this information in the manuscript in result section line number 280 to 289. 

Comment 3. Fold change analysis of FT-IR data shows significantly altered peaks at 3300, 2950, 1645, 1540 and 1448 cm-1 between asymptomatic and acute cases. What about the altered peaks in chronic cases? The authors need to discuss this point.

Ans- FTIR peaks at 3300, 2950, 1645, 1540 and 1448 cm-1 were significantly altered in asymptomatic and acute case. However, in chronic case these peaks were not altered significantly as compared to control. This difference in chronic and acute/asymptomatic samples could be either due to absence of microfilariae in chronic patients or due to change in immune response after progression to chronic stage. We have added this information in the manuscript in discussion section line number 418 to 425.

Comment 4. The authors claim that a combination of different techniques (FTIR, MMP zymography, SDS-PAGE, 2DGE, MALDI-TOF/MS etc.) was used to identify LF biomarkers from serum samples of different stages of LF patients. How do the authors conclude that the biomarkers thus detected were due to LF infection only and not due to any other previous ailment or other diseased conditions? Merely writing, “LF infected cases were examined by a clinician and were categorized based on the above mentioned manifestations and presence/absence of microfilariae in the bloodstream” does not hold much value as patients exhibiting Tropical Pulmonary Eosinophilia do not show presence of Mf in the peripheral blood. What was the criteria used for exclusion by the clinician, needs to be mentioned in detail.

Ans- In the present study well characterized LF serum were used. The LF patients were examined by the clinician and expert medical doctors. Only those who were not suffering from TPE were included in this study. We have also added this information under materials and methods line number from 163 to 170 as Inclusion and Exclusion criteria.

Comment 5. Can the adopted approach be used to distinguish LF infection caused by Wuchereria bancrofti, Brugia malayi and Brugia timori? This needs to be discussed by the authors.

Ans- We cannot comment on this at the moment because we have not included B. malayi and B. timori so far in our study. In future we can include Brugian samples also.

Comment 6. What was the duration of the study? Since acute infection cases would progress to chronic cases, won’t their serum profile differ over a period of time and match with the chronic cases? What were the time points of blood withdrawal? Single time point withdrawal is not sufficient to provide the complete picture as gravid females would continuously produce Mf which may alter serum profile? Also, age and gender would make a difference.

Ans- The study was done from March 2019 to July 2021. The blood samples were collected at different time and blood collection was done for almost 3 years for this study. The mixed male and female population was chosen between 20 to 60 year age group. We have taken blood samples from every patient only once, after which they were administered DA and continuous treatment and monitoring was done as per WHO guidelines. 

The purpose of this study was to identify potential markers for LF diagnosis which are not age and gender specific. However, in future we plan to extend this study of LF markers using antibody based assay wherein age and gender can be taken into account.

Comment 7. Zymography and MMP data is not very convincing, as very small differences exist between groups. When one takes SD into account, there would be hardly any differences

Ans- In Supplementary table- 2 the image intensity along with fold change and p value is provided for bands of 240, 72 and 92 Kda. The significantly altered bands in MMP zymography are denoted with asterisk for p value > 0.05. The quantitative analysis for fold change of all the gel images was done by Quantity one and Image J software.

Comment 8. Fig 2, Marker well has different contrast (top and bottom) how come?

Ans – Now, a clear image is provided in figure 2.

Minor concerns:

Concerns 1. The authors need to factually correct the starting line of abstract and introduction. As per latest WHO figures, 863 million people are affected by LF, the number citing 120 million infections is a very old data (https://www.who.int/news-room/factsheets/detail/lymphatic-filariasis).

Ans: Corrected.

Concerns 2. Under “Separation of immune complexes”, either the authors must mention the name of author/research group or write “as described previously”. Just writing “LF cases were isolated by the method of [15] with minor modification” is not correct.

Ans Corrected.

Concerns 3. Length of IPG strips and pH gradient used in 2DGE should be included in M & M, rather than just in the results.

Ans Done.

Concerns 4. Fig 3b, y axis should be “change”

Ans- Done.

Concerns 5. Clearly state statistical comparisons have been made between which groups in fig 1 a and 1b?

Ans – Done and highlighted in yellow color in line number 293 to 297.

Concerns 6. In Abbreviation and abstract, GPELF should read as Global Programme to Eliminate Lymphatic Filariasis.

Ans – Corrected.

---

## [Decision Letter · Decision Letter 1]

15 Jun 2022

Lymphatic Filarial Serum Proteome Profiling for Identification and Characterization of Diagnostic Biomarkers

PONE-D-22-04328R1

Dear Dr. Singh,

We’re pleased to inform you that your manuscript has been judged scientifically suitable for publication and will be formally accepted for publication once it meets all outstanding technical requirements.

Kind regards,

Suprabhat Mukherjee, Ph.D.

Academic Editor

PLOS ONE

Additional Editor Comments (optional):

Authors have addressed all the comments/concerns and the revised version is acceptable for publication.

Reviewers' comments:

Reviewer's Responses to Questions

**Comments to the Author**

1. If the authors have adequately addressed your comments raised in a previous round of review and you feel that this manuscript is now acceptable for publication, you may indicate that here to bypass the “Comments to the Author” section, enter your conflict of interest statement in the “Confidential to Editor” section, and submit your "Accept" recommendation.

Reviewer #1: All comments have been addressed

Reviewer #2: All comments have been addressed

2. Is the manuscript technically sound, and do the data support the conclusions?

Reviewer #1: Yes

Reviewer #2: Partly

3. Has the statistical analysis been performed appropriately and rigorously? 

Reviewer #1: Yes

Reviewer #2: No

4. Have the authors made all data underlying the findings in their manuscript fully available?

Reviewer #1: Yes

Reviewer #2: No

5. Is the manuscript presented in an intelligible fashion and written in standard English?

Reviewer #1: Yes

Reviewer #2: Yes

6. Review Comments to the Author

Reviewer #1: (No Response)

Reviewer #2: Most of the comments have been addressed. However, I still see that minor concern 4 has not been addressed, and minor concern 5 has not been responded properly.

7. PLOS authors have the option to publish the peer review history of their article (what does this mean?). If published, this will include your full peer review and any attached files.

Reviewer #1: **Yes: **Niladri Mukherjee

Reviewer #2: No

---

## [Editor Report · Acceptance letter]

23 Jun 2022

PONE-D-22-04328R1 

Lymphatic Filarial Serum Proteome Profiling for Identification and Characterization of Diagnostic Biomarkers 

Dear Dr. Singh:

I'm pleased to inform you that your manuscript has been deemed suitable for publication in PLOS ONE. Congratulations! Your manuscript is now with our production department. 

Kind regards, 

on behalf of

Dr. Suprabhat Mukherjee 

Academic Editor

PLOS ONE